# The Effect of Different Substrates on the Morphological Features and Polyols Production of *Endomyces magnusii* Yeast during Long-Lasting Cultivation

**DOI:** 10.3390/microorganisms10091709

**Published:** 2022-08-25

**Authors:** Anastasia S. Kokoreva, Elena P. Isakova, Vera M. Tereshina, Olga I. Klein, Natalya N. Gessler, Yulia I. Deryabina

**Affiliations:** 1A.N. Bach Institute of Biochemistry, Research Center of Biotechnology of the Russian Academy of Sciences, Leninsky Ave. 33/2, 119071 Moscow, Russia; 2Winogradsky Institute of Microbiology, Research Center of Biotechnology of the Russian Academy of Sciences, Prospekt 60-Letiya Oktyabrya, 7/2, 117312 Moscow, Russia

**Keywords:** *Endomyces magnusii*, mitochondria, yeast, long-lasting cultivation, cytosol carbohydrates

## Abstract

The study on the influence of different glucose concentrations (2%, 0.5%, and 0.2%) and glycerol (1%) on the morphological and physiological features, as well as the composition of soluble carbohydrates, was performed using *Endomyces magnusii* yeast. Two-factor analysis of variance with repetitions to process the data of the cell size changes showed that the substrate type affected cell size the most. The cells with 2% glucose were 30–35% larger than those growing on glycerol. The decrease in the initial glucose concentration up to 0.5–0.2% slightly changed the cell length. However, even in the logarithmic growth phase pseudo-mycelium of two to four cells appeared in the cultures when using low glucose, unlike those using glycerol. Throughout the whole experiment, more than 90% of the populations remained viable on all of the substrates tested. The ability for colony formation decreased during aging. Nevertheless, at the three-week stage, upon substrate restriction (0.2% glucose), it was twice higher than those under the other conditions. The respiration rate also decreased and exceeded not more than 10% of that in the logarithmic phase. By the end of the experiment, the cyanide-sensitive respiration share decreased up to 40% for all types of substrates. The study of soluble cytosol carbohydrates showed that the cultures using 2% glucose and 1% glycerol contained mainly arabitol and mannitol, while at low glucose concentrations they were substituted for inositol. The formation of inositol is supposed to be related to pseudo-mycelium formation. The role of calorie restriction in the regulation of carbohydrate synthesis and the composition in the yeast and its biotechnological application is under consideration.

## 1. Introduction

Various harmful changes accumulate when any organism becomes old, which decrease its resistance to diseases, resulting in death, and is the major reason for aging. The free radical theory of aging was declared in the 1950s [1]. It was postulated that the reactive oxygen species (ROS) together with some endogenous metabolic reactions have negative effects and lead to the aging and death of all living things [2]. Senescence is supposed to be related to the increased ROS generation, while the scavenging cellular system activity declines. Both impacts cause crucial damage of the mitochondrial action and cellular physiology in general. An increase in the metabolic process activity (largely, the activity of the respiratory enzymes) enhances the action of free radical reactions (FRR), which increases the intracellular ROS levels, promoting not only the aging processes, but also the life evolution, genetic defects, and different diseases of organisms [2]. The mitochondrial respiratory chain, non-enzymatic reactions involving oxygen, phagocytosis, prostaglandin synthesis, cytorochrome P-450 system, and ionizing radiation (in mammals) are the sources for FRR. The FRRs intensity and the accumulation of ROS-induced disorders in the cell increases with age, which, in its turn, leads to an “age-related” depression of cell functions and, finally, to the death of the organism [1].

Calorie restriction (CR) is a diet, when only the calorie intake is limited, while the intake of any other nutrients (amino acids, vitamins, and trace elements) is not. CR could mitigate the negative effect of age-related changes at the cellular level [3]. It is the most studied and reproducible non-genetic intervention, which increases the life-span and improves the health of organisms at various organization levels, from unicellular yeast to primates and humans. It indicates the involvement of some conservative mechanisms being universal for all eukaryotes in the process. It has been confirmed by numerous studies [4,5]. Conservative processes including autophagy, TOR signaling, and AMP-activated protein kinase (AMPK) signaling participate in the regulation of aging in most CR models [6].

Yeast organism, in particular the baker’s yeast of *Saccharomyces cerevisiae* is a successful model of unicellular organisms to study aging processes and their correction [7,8,9]. Two aging models have been studied using yeast organisms: the number of daughter cells, which a mother cell produces before physiological aging, determines the replicative life-span (RLS); the chronological life-span (CLS) defines the period in which a cell lives in a non-dividing state [8,10]. To obtain the calorie restriction effect in yeast, either the glucose concentration (0.5% and less) or the total amino-acids amount is decreased [11,12,13,14]. Calorie restriction is usually considered to slow down cellular metabolism, thereby reducing oxidative stress [15]. Limiting the glucose concentration in the culture medium was recorded in order to prolong both RLS and CLS [11,16,17,18]. However, some studies suggest that the benefit to life-span should involve the activation of the regulatory protein SIR2 (Sir2p) by NAD as a specific molecular mechanism. In yeast, Sir2p induction could extend their life-span through the increase in silencing and rDNA stability. As for metazoans, as in yeast, Sir2p is likely to promote genomic silencing as its mammalian homolog, mSir2a, is also an NAD-dependent histone deacetylase. Perhaps a silencing support is crucial to longevity in metazoans both repressing genomic instability and preventing inappropriate gene expression [19]. The increase in the CLS is related to the suppression of the nutrient signaling pathways, such as the conserved TOR and Ras/cAMP/PKA pathways, which regulate cell growth in response to nutrient availability [20]. This leads to the activation of Rim15, which, in turn, induces transcription factors Msn2/4 and Gis1, contributing to the transition to the stationary phase, the trehalose and glycogen accumulation, followed by an increase in stress resistance [21]. At the same time, in the yeast cells, CR promotes an enzymatic defense system against ROS [22]; however, the number of the mitochondria remains the same [23].

The study of yeast cells has definitely shown that ROS accumulate during the so-called diauxic transition (the transition to the stationary growth phase) and the stationary growth phase [24]. It has also been reported that nutrient limitation in the yeast population, particularly of glucose and amino acids, triggers an increase in both RLS and CLS. As the glucose level decreases at the end of the exponential growth phase, the cells switch from the enzymatic metabolism (glucose conversion into ethanol) to the mitochondrial oxidative metabolism of ethanol and organic acids produced during the cultivation. This “diauxic shift” is accompanied by an abrupt change in transcription, translation, and metabolic profile, resulting in a slower culture growth and, finally, entry into the quiescence (G_0_ phase) [25,26]. Oxidative metabolism in the stationary growth phase affects not only carbohydrate metabolism, but also lipid synthesis, ROS generation, mitochondria activity, stress response, and apoptotic reactions [27,28,29].

Previously, we studied the metabolic profile of the obligate aerobic yeast of *Endomyces magnusii*, which possesses a complete respiratory chain and a well-developed mitochondria system using glycerol and glucose as carbon sources [30]. The *E. magnusii* yeast, being an obligate aerobe, possesses the complete respiratory system in its mitochondrial system, similar to the animal ones that makes it a unique model for studying some mechanisms including the ageing. The membrane and mitochondrial apparatus of the yeast with the fermentative metabolism type, for example *S. cerevisiae*, is ill-developed, and there are only scanty large mitochondria with small cristae in the cell. Their mitochondria have much fewer cristae and trend to irregularity in their shape, structure, and packing. However, the yeast of clearly-pronounced aerobic metabolism, namely *E. magnusii*, possess a well-developed membrane apparatus with abundant complex mitochondria with numerous cristae. The composition of the mitochondria respiratory chain in traditional yeast species such as *S. cerevisiae* is quite variable. The *E. magnusii* yeast possesses invariable complex I and the absence of alternative pathways of respiration [31]. Our results revealed an increased level of ROS, activities of antioxidant systems (catalase, superoxide dismutase, and the glutathione system), and a high cellular redox potential in the late and deep stationary growth phases in the culture growing on glycerol. Moreover, we observed high survival under those conditions, while in the culture growing on glucose, its survival significantly decreased by the seventh day of growth. Our data confirm the hypothesis that ROS, generated in eukaryotic mitochondria during oxidative phosphorylation, play a key role in regulating life-span at all levels of life organization, from yeast to humans [32]. According to the hypothesis, the Kreb’s cycle intermediates, namely α-ketoglutarate, and the mitochondrial ROS regulate the transcription factor activity, for example, Hif1, leading to a significant readjustment of the cellular metabolism. However, the key metabolic pathways affect ROS levels and antioxidant signaling, thereby mutually regulating life-span. We concluded that after the growth of *E. magnusii* yeast on glycerol, using calorie restriction conditions, there was an increase in H_2_O_2_ production at the stationary growth stage that promoted a rise in SOD activity and prolonged CLS under those conditions. This phenomenon is the result of the hormetic effect of ROS produced by the mitochondrial electron transport chain, and explains the high activity of antioxidant protective enzymes and the glutathione system in combination with the NADPH synthesis system, which occurred in the experiments with nutritional restriction.

We concluded that at the stationary growth stage, the growth of the *E. magnusii* yeast on glycerol modeling the conditions of caloric restriction led to an increase in H_2_O_2_ production, which promoted SOD activity and prolonged CLS under those conditions. This phenomenon is the result of the hormetic effect of the ROS generated in the mitochondria and it can explain the high activity of the antioxidant protective and glutathione systems when combined with the NADPH synthesis system occurring after nutritional restriction. Although the assumption about hormesis partially contradicts the well-known theory by Harman, it sheds some light on the data obtained in our study and in those by some other researchers [33].

In this study, we performed a wide-ranging analysis of the CR effect on the morphometric features, energy status, and metabolic profile of *E. magnusii* cells.

## 2. Materials and Methods

### 2.1. Yeast Strains and Growth Conditions

The *E. magnusii* yeast strain VKM Y261 received from the collection of the G.K. Scriyabin Institute of Biochemistry and Physiology of Microorganisms of the Russian Academy of Sciences (IBPM RAS) was raised in 100 mL of glycerol- (1%) and glucose (0.2%; 0.5%; 2%)-containing media. The composition of the medium was as follows (g/L): CaCl_2_—0.05, MgSO_4_—0.5, KH_2_PO_4_—8.6, NaCl—0.1, (NH_4_)_2_SO_4_—0.3, yeast extract 2.0, *L*-tryptophan 2.75 mg, *L*-methionine 2.75 mg, and *L*-histidine–2.75 mg at 28–29 °C, as it was described before [30]. The cellular suspension was assessed using absorbance at a wavelength of 590 nm (A_590_) with a Specol-11 spectrophotometer (Carl Zeiss, Oberkochen, Germany). The cells were raised in various growth phases, namely, the exponential (18.5–19 h of growth, A_590_ = 2.6–2.7), early stationary (48 h of growth, A_590_ = 4.5–4.6), late stationary 1 (1 week of growth, A_590_ = 4.4–4.7), deep stationary 1 (2 weeks of growth, A_590_ = 4.4–4.7), deep stationary 2 (3 weeks of growth, A_590_ = 4.4–4.7), and deep stationary 3 (4 weeks of growth, A_590_ = 4.4–4.7) stages.

### 2.2. Cell Viability and Vitality Assays

To assay the yeast viability, the cells of the stationary stage were centrifuged, washed using sterile water, and re-suspended in a 100 mM KPi buffer to a final cell concentration of 10^8^ cells mL^−1^; pH 7.0. To assess cell viability and vitality, some independent methods were used [34].

#### 2.2.1. Spotting Test

The culture was centrifuged, re-suspended in distilled water, and diluted to final densities of 10^5^, 10^4^, or 10^3^ cells mL^−1^. The aliquots (10 µL) of each dilution were plated on a solid YPD medium and incubated at 29 °C. Colony growth was inspected after 48 h.

#### 2.2.2. Staining with Methyl Blue

The yeast cultures were centrifuged, suspended using phosphate-buffered saline (PBS), and added to a 200 µL aliquot of 100 µL methylene blue (0.1 mg mL^−1^ stock solution, dissolved in a 2% dihydrate sodium citrate solution), and the samples were incubated for 5 min at room temperature. The survival rate of about 1000 cells in one biological replicate was assayed using Gorjaev’s chamber (×400) under a light microscope. Live cells were colorless, and dead ones were blue.

#### 2.2.3. CLS Assay

The samples of the cultures at various growth stages were diluted to assay the total number of cells using microscopy. Serial dilutions of the yeast cultures were inoculated on solid YPD (1% yeast extract, 2% peptone, 2% dextrose, and 2% agar) medium in two replicates to count the number of live cells in each variant. The cultures raised using 1-% glycerol and 2% glucose were washed and suspended in distilled sterile water, then diluted up to a final density of 10^6^ and 10^7^ cells mL^−1^. For the cells using 0.2 and 0.5% glucose, the dilution was of 10^5^ or 10^4^ cells × mL^−1^. The plates were kept at 29 °C for 24–48 h. We calculated the number of colony forming units (CFU) as the number of reproductively capable cells in a sample as follows: number of CFU × dilution × factor × 10 = number of reproductively capable cells per mL. The colony growth was inspected after 48 h, and the number of grown colonies were counted on each plate. The total number of cells was assayed under a light microscope using Gorjaev’s chamber (×400) on at least 1000 cells in one biological replicate.

#### 2.2.4. Flow Cytometry

The cells were centrifuged and resuspended in a medium to determine the cell concentration and viability using propidium iodide (PI). For the viability assay, PI staining and acquisition were performed as follows. Cells (1 mL) were transferred into a 15 mL conical tube, centrifuged at 3000 g for 10 min, and resuspended in 100 μL of PBS. After two washes in PBS, the cells were resuspended in 1.5 mL PBS and divided into three aliquots of 500 μL each (negative control, positive control, and a sample tested). The positive control was heated up to 100 °C for 5 min. Then, 0.5 μL PI was added to the cells of the positive control and the sample was stirred and then incubated at 37 °C for 30 min. Acquisition was performed on a cytometer CytoFLEX^®^ (Beckman Coulter, Brea, CA, USA). For the cells stained with PI, the maximum fluorescence excitation was used of 535 nm (green laser), with a maximum emission of 617 nm (red channel). The data obtained were analyzed using the CytExpert software v.2.4.

### 2.3. Cell Respiration

The consumption of oxygen of the yeast cultures was assayed in vitro at +25 °C polarographically with electrodes covered by a fluoroplastic film at a constant potential of 660 mV. The medium was composed of 50 mM KPi, pH 5.5, with 1% glucose applied [35].

### 2.4. Carbohydrate Analysis

The carbohydrates were extracted with extremely hot water for 20 min, four times. From the resulting extract, we removed the proteins and then purified the charged substances using a column with both the Dowex-1 (acetate form) and Dowex 50 W (H^+^) ion exchange resins [36]. The carbohydrate composition was assayed using gas–liquid chromatography with trimethylsilyl derivatives, which were obtained from the lyophilized extract [37]. We used *α*-Methyl-*D*-mannoside (“Merck”, Kenilworth, NJ, USA) as the internal standard. Gas chromatography was performed with a Kristall 5000.1 gas chromatograph (“Chromatek”, Yoshkar-Ola, Russia) equipped with a ZB-5 30 × 0.32 mm capillary column (“Phenomenex”, Torrance, CA, USA). The temperature program was set at +130, 5–6 °C/min gradient to +270 °C. We used the markers of glucose, mannitol, arabitol, inositol, and trehalose (“Sigma”, St. Louis, MO, USA) as the standards.

### 2.5. Statistical Analyses

The presented data were calculated as the average ± the standard deviation in biological triplicates with a standard error of less than 5%. Statistical analysis was performed using the method of two-factor analysis of series with repetitions (*n* = 100). We used graphical and computational methods to test the normal distribution for the cell length data. Calculation methods included the compared mean values of the sample and its median and the calculated Kolmogorov–Smirnov index. For all of the series, the mean and median were approximately equal, and *p* was more than 0.05 according to the Kolmogorov–Smirnov test. The results indicate that the distribution of all of the series was normal. An analysis of the soluble carbohydrates and lipids was performed using one-way ANONA (*n* = 3). *p* values were determined by the two-tailed paired *t*-test at a 5% level of probability.

## 3. Results

### 3.1. Growth Features of the E.magnusii Yeasts upon the Cultivation Using Different Substrates

The *E. magnusii* yeast is a kind of poly-nuclear fungus (Figure 1A) that is able to form the pseudomycelium and the true mycelium (Figure 1C,D). The *E. magnusii* yeast, being a definite aerobe, has a well-developed membrane with numerous mitochondria with exuberant cristae (Figure 1B). The growth of the *E. magnusii* cultures differed a lot due to the various types and concentrations of the substrates studied (Figure 1E). Being grown on an oxidative substrate of glycerol, the culture accumulated biomass for about 20 h up to the stationary growth stage and continued growing during that phase, which lasted for 16 h, i.e., for a total of 36 h (Figure 1E). Then, the biomass accumulation rose a bit at the 36–53 h growth stage, followed by constant stationary growth during the whole experiment, and at the 168 h (1 week) growth stage, a 20% transitive decrease was observed (Figure 2E). The experiment was performed for more than for 4 weeks and there were no significant shifts in the stationary stage (data not shown). The growth pattern was quite different using the “fermentative” substrate (2% glucose). The maximum biomass yield increase for the 1% glycerol was about 13–16% lower than that for the 2% glucose (Figure 1E), while the stationary phase was much shorter (no longer than 40 h) and was accompanied by no change in the biomass concentration, with a slight decrease at the 50–672 h (4 weeks) stage (Figure 1E). The graph shows that the culture grew best on the 2% glucose- and 1% glycerol-containing medium. At low glucose concentrations (0.5% and 0.2%), the biomass yield was significantly lower and reached only 55% and 36% of the growth using glycerol, and 48% and 32% of the growth using 2% glucose, respectively (Figure 1E). When low concentrations of glucose were utilized, the culture concentration dropped during the 168 h (1 week) stage by 8% and 26% in the case of the 0.5% and 0.2% glucose, respectively (Figure 1E).

### 3.2. The Cell Morphology upon Long-Lasting Cultivation Using Different Substrates

The microscopy of the *E. magnusii* cells that were increased using different substrates displayed polymorphism of the population even at the logarithmic growth stage. The cells raised using 1% glycerol were oval, ovoid, or cylindrical, and were 6–30 µm long and 6–10 µm wide (Figure 2A(a)). The cells growing on glycerol were about 1.5 times smaller than those on 2% glucose (Figure 2B(b)). Most cells using 2% glucose were elongated, cylindrically shaped, and up to 42 µm long and 16 µm wide (Figure 2B(b)). As the population was aging, the cell length decreased. The cells grown using 0.5% glucose were elongated, cylindrical, seldom oval, 12 to 44 µm long, and 5 to 16 µm wide (Figure 2C(c)). Even at the logarithmic stage, some filamentous cells appeared. By the fourth week of growth the share of those structures reached 21.6%. The dynamics of size changes for the cells raised on 0.2% glucose were similar to those on the 0.5% glucose. However, the share of the filamentous cells reached 47.7% at the two-week stage (Figure 2D). It is of interest that in all of the versions cultivated, at the 4 week stage of growth, either dividing (Figure 2A,B) or budding cells (Figure 2C,D) were observed. The cells size did not differ so significantly during cultivation. However, the populations raised on different substrates varied alot in their size. The cells using glycerol were nearly two-fold smaller than those growing on glucose. Upon using glycerol, the cells never formed a filamentous form.

We used graphical and computational methods to test the normal distribution for the cell length data. Statistical processing was performed using a method of two-factor analysis of variance with repetitions (*n* = 100). Calculation methods included the compared mean values of the sample and its median, and the calculated Kolmogorov–Smirnov index. For all the series, mean and median were approximately equal, and *p* was more than 0.05 according to the Kolmogorov–Smirnov test. The results indicate that the distribution of all the series was normal. The results indicated that both the substrate and age, together with their interaction, affect the cell length. Calculating the ratio of the share of the square sum of the factor tested to the sum of all the squares of the variances. The substrate affected cell length the most (34.6%) and age less (5.3%), and the interaction of those factors (2.5%) decreased the effect.

Thus, the morphology of *E. magnusii* cells mainly depends on the substrate and its concentration. The cells grown using 0.5% and 2% glucose were more elongated and they were, on average, 1.85 times longer compared with those grown on glycerol (according to the cell medians) (Figure 2B,C,E). The cells grown using 0.2% glucose varied most of all during the cultivation and the scatter reached 22%. Moreover, the maximum was at the two-week stage, and it was similar to those grown using 0.5 and 2% glucose (Figure 2D–F). Numerous mycelial structures in the populations grown using low glucose (0.5% and 0.2%) was another remarkable difference (Figure 2C,D). This is because, in the cultures grown on high amount of substrates, mycelium forms could hardly occur as single cells at the late growth stages of growth (2–4 weeks).

### 3.3. The Dynamics of the Survival of the E. magnusii Yeast during the Long-Term Cultivation Using Different Substrates

The survival rate of yeast cultures was assayed using methods of cell staining with methylene blue, flow cytometry, and colony forming capability. The cells grown on 0.2% glucose (92.8% and higher) displayed the highest survival rate throughout the whole experiment (Figure 3A). The survival rate was slightly lower in the cell cultures, which grew in the 1% glycerol- and 0.5% glucose-containing media (Figure 3A). So, when cultivating on 1% glycerol, the survival rate was 100% at the one-week stage, and with aging it decreased by 13%. The survival rate of the culture using 0.5% glucose was 97% at the one-week growth stage, and then decreased by about 11% (Figure 3A). The cells grown on 2% glucose showed the lowest survival rate in the one-week stage. In the one-week growth stage, the survival rate was 87.7%, which decreased by 15% while getting older (Figure 3A).

Upon long-lasting cultivation, the *E. magnusii* yeast colony forming ability decreased using all the substrates (Figure 3B). The cultures grown using both 2% and 0.5% glucose had similar results for the CFU at the two-week stage and by the three-week stage, in those cells, it nearly halved, decreasing at the four-week stage nearly seven-fold in the cells grown on 0.5% glucose. The cells grown at 0.2% for three weeks showed the highest colony forming ability (Figure 3B). So, in the first-week stage, it was 62%, which decreasing after three weeks to 40%. At the four-week stage, it declined to 14%. Thus, the CFU declined four-fold by the end of the experiment. As for the population grown on 1% glycerol, at the four-week stage, its CFU was nearly twice as high as that in the population grown on glucose (Figure 3B).

Flow cytometry using PI stained cells indicated a high viability for all of the cell populations both at the stage of logarithmic growth and throughout the whole experiment. Moreover, the number of cells that were not stained with PI reached more than 95% for the populations on both glucose and glycerol (Appendix A). It is noteworthy to say that the cells grown on glycerol at the logarithmic growth stage separated into two populations: large cells (~65%) and small cells (~35%) (Appendix A). About 99% of the whole population were negatively stained with PI (Appendix A), and 98% of the cells grown on 2% glucose contained small cells (Appendix A). About 99% grown on glucose were out of the positive control zone (Appendix A). In the logarithmic growth stage, the cells grown on 0.5% glucose were separated into two populations: the first one containing large cells (nearly 70%) and the second one with small cells (~30%) (Appendix A). Both the large and small cells showed a high survival rate, namely more than 99% of the cells showed a high survival rate. In the logarithmic growth stage, the cells grown on 0.2% glucose also separated into two populations of large cells (~67%) and small cells (~33%) (Appendix A). Both types of cells showed high survival rates. However, small cells showed a survival rate of about 95% (not shown), while the populations of large cells showed nearly 99% (Appendix A). The cells grown on glycerol for 2 weeks formed one population of large cells (Appendix A). Live cells made up more than 99% of the cells grown on glycerol. The population of the cells grown on 2% glucose were stratified into the part of large cells (96%) and that of long cells (4%) (Appendix A). Both the large and long cells had more than a 99% survival rate (Appendix A). At the stationary growth stage, the cells grown on 0.5% glucose contained predominantly large cells (96%) with a small population of small cells (4%) (Appendix A). Live cells made up 99% of large cells (Appendix A) At the same time, only 82% of small cells were alive (not shown). Perhaps, the observed effect is related to the worse permeability of small cells for PI. At the stationary growth stage, the population of cells grown on 0.2% glucose contained predominantly large cells (95%) with a small portion of small cells (4%) (Appendix A). Live cells made up 99% of large cells (Appendix A). However, only 77% of small cells were alive (not shown). In 4 weeks of growth, the cells using glycerol formed a relatively homogeneous population (Appendix A). More than 99% of its cells were alive at the 4-week stage (Appendix A). When the cells were cultivated on 2% glucose for 4 weeks, the cells were split into two populations: the minor and major ones (Appendix A). More than 99% of the cells were live. The cells cultured on 0.5% glucose formed a homogeneous population (Appendix A). At the four-week stage, more than 98% of the cells were quite viable (Appendix A). However, staining of the cells with PI showed a bimodal distribution, namely about half of the cells were more permeable to PI. In the case of 0.2% glucose, at this stage, the cells formed a dense population, 98% of which showed a high survival rate (Appendix A).

Thus, the survival rate obtained with methylene blue or PI staining cells remained very high throughout the whole experiment using all of the substrates tested. Comparing those data with the results using the CFU method, we conclude that at the later stages of long-lasting cultivation, the populations mainly consisted of quiescent cells incapable of proliferation and colony formation. Under the conditions of caloric restriction (0.2% glucose and 1% glycerol), the cultures maintained a higher colony-forming capability during the whole experiment.

### 3.4. The Respiratory Features of the E. magnusii Yeast upon Long-Lasting Cultivations on Different Substrates

The respiration rate (Figure 4) in the cells grown using all of the substrates was the highest in the logarithmic phase of growth and halved by the third or fourth day of cultivation (data not shown). By the end of the experiment, the respiration rate exceeded not more than 10% of the initial one (Figure 4). The maximum respiration rate in the cells grown on 2% glucose was observed at the end of the logarithmic growth phases. Then, it decreased nearly two-fold and remained at this level up to the two-week stage, and then gradually decreased up to 10% in the case of glycerol and 2% glucose, but up to 3.75% and 7.8% in the case of 0.5% and 0.2% glucose, respectively (Figure 4). The inhibition by cyanide in the logarithmic and the beginning of the stationary growth stages reached nearly 90%, but by the fourth day, it decreased up to 67% and remained at this level throughout the whole experiment (Figure 4).

In the cells using 0.5% glucose, the respiratory rate was at a maximum in the logarithmic and early stationary growth phases, and exceeded that for the 2% glucose by nearly one and a half times. At the one-week stage, the respiratory rate gradually decreased during the cultivation. The dynamics of cyanide inhibition were similar to that on 2% glucose (Appendix A). When being cultivated on 0.2% glucose, in the stationary phase, the respiration rate was comparable to that for the cells grown on 0.5% glucose, and then it declined more rapidly (Figure 4). The cyanide inhibition in the logarithmic phase reached up to 87.5%, and then it decreased to 70% (Appendix A).

In the logarithmic growth phase, the respiratory rate of the cells grown on glycerol was similar to those using 0.5 and 0.2% glucose, and a bit higher than those in the cells using 2% glucose (Figure 4). In general, the dynamics of the respiratory rate changes during long-lasting cultivation were similar on all of the substrates tested. By the end of cultivation, the respiration rate exceeded no more than 10% of the initial one (Figure 4). The cyanide respiration inhibition in the logarithmic growth phase reached 90–100% on any substrate, with the lowest level in the cells using 0.2% glucose (87.5%) (Appendix A). During cultivation, the cyanide-resistance of the culture increased up to about 30% with the highest level in the cells using 1% glycerol (35%) (Appendix A).

### 3.5. The Dynamics of Soluble Carbohydrate Profile Changes in the E. magnusii Yeast upon Long-Lasting Cultivation on Different Substrates

The total amount and composition of soluble carbohydrates changed significantly during long-lasting cultivation using different substrates. Figure 5A shows that upon cultivation using 2% glucose, the soluble carbohydrates content was the lowest at the one-week stage. In the 0.5% glucose-containing medium, after the first week, the soluble carbohydrates level in the cytosol was four to six times higher than that during further cultivation. The cells grown using 0.2% glucose possessed the lowest level of soluble cytosol carbohydrates throughout the whole experiment (0.3–0.6% of dry biomass). The level of soluble carbohydrates in the cytosol in the cells grown using 1% glycerol increased by 25-fold at the two-week stage compared with that at the one-week stage and remained at the highest level among the other versions throughout the whole experiment.

In the soluble carbohydrates of *E. magnusii* (Figure 5B), arabitol, mannitol, and inositol were dominant. Trehalose, glucose, and glycerol, as the minority, exceeded no more than 6% of the total amount. No erythritol was detected. The cells grown on 2% and 0.5% glucose contained mainly arabitol (93 and 83%, respectively) at the one-week stage (Figure 5B). Small amounts of glucose, glycerol, inositol, and mannitol were detected. In the cells grown on 0.2% and 0.5 glucose, the inositol level reached about 60% of the soluble carbohydrates (Figure 5B). Glucose, mannitol, and trehalose were also observed. Arabitol made up about 13%. In the cells grown with 1% glycerol, arabitol was about 70%, and for glycerol and glucose (more than 8%), inositol (about 7.5%) and trehalose (more than 3%) were also found (Figure 5B). No mannitol was revealed. Aging of the populations on 2% glucose and 1% glycerol increased the mannitol share up to 20% at the expense of some other sugars. Arabitol remained the main polyol. In the cells grown on 0.5% and 0.2% glucose, the inositol fraction increased to 90% or more, while the arabitol share significantly decreased (Figure 5B).

Thus, the polyols of arabitol, mannitol, and inositol were the main soluble carbohydrates. Moreover, arabitol and mannitol were more common for the cells using 2% glucose and 1% glycerol, but in the cells grown on 0.2% glucose, inositol was dominant. The cells growing using a concentration of 0.5% glucose occupied the intermediate position. At the one-week stage, arabitol and mannitol dominated in those cells, which is typical for cells using 2% glucose. However, in aging cells, inositol was substituted, finally reaching about 80% of the total amount, which is typical of the cells growing on 0.2%. In the media with low glucose concentrations, inositol formation induced mycelium formation.

## 4. Discussion

In the present study, we tried to thoroughly analyze the effect of calorie restriction on the morphometric and biochemical parameters of an aging *E. magnusii* culture during the 4-week cultivation. We found some interesting facts. (1) When the concentration of the fermentable substrate (glucose) was limited to 0.5–0.2%, mycelial forms appeared and the cell morphotypes separated into three groups—dividing cells (buds), single ones (yeast, vegetative forms), and mycelial ones (Figure 2C,D). (2) Arabitol and mannitol were the main soluble cytosolic carbohydrates in the cells using 2% glucose and 1% glycerol, while the cells growing on 0.2% glucose contained mainly inositol, and the cells growing on 0.5% glucose, which contained both inositol and arabitol, occupied the intermediate position (Figure 5B).

These experimental data are supposed to be interrelated. Thus, in *S. cerevisiae* yeast, it was found that soluble inositol polyphosphate is able to act as the signaling molecules needed for the dimorphic transition from vegetative to pseudo-hyphal growth to form pseudomycelium [38]. The transition from unicellular yeast to the formation of multicellular filaments is known to be typical for many fungi [39,40,41,42]. In the budding cells of *S. cerevisiae*, either the restriction of nitrogen sources or growth on alternative carbon sources causes cell elongation, unipolar budding, and altered intercellular adhesion leading to the formation of pseudo-hyphal filaments, which seems to benefit a unicellular organism [43,44,45]. Gimeno et al. [43] showed that upon limitation in nitrogen, *S. cerevisiae* diploid strains undergo a dimorphic transition, changing their shape and division pattern, which leads to the appearance of invasive filamentous forms. The cells become longer and thinner, forming pseudo-hyphae, which grow from the colony and penetrates the solid medium. The filamentous growth is supposed to allow a yeast cell to receive nutrients more completely. Such a type of growth demands only polar budding of diploid cells, while no dimorphic transition is observed in their haploid axially budding cells. The activated RAS2 or mutation in SHR3, which is needed for the uptake of amino acids, enhances pseudohyphal growth. Nevertheless, the dominant mutation, which causes random budding, blocks this kind of growth [43]. Classical studies have identified a core set of conserved signaling modules, which could regulate pseudo-mycelium growth in the yeast, namely the Kss1p mitogen-activated protein kinase (MAPK) cascade, the Snf1p family of AMP-activated kinases, and the Ras2p/cAMP-dependent protein kinase A (PKA) pathway [45]. Genomic studies using knock-out mutants and overexpression libraries have shown a much wider set of genes needed for pseudo-mycelium growth [46,47,48], but the changes in metabolite levels underlying the transition to filamentous growth still remain unknown.

Apparently, under conditions of a low concentration of carbohydrate substrates, the formation of inositol in the *E. magnusii* yeast may also be necessary to form pseudomycelial and true mycelial structures. Its share at the four-week stage of cultivation using 0.2% glucose reached 60% (Figure 2D,E). Using the statistical analysis, we showed that the substrate mostly affected cell length (34.6%) and less the age (5.3%), and the interaction of these factors (2.5%) had the least effect (Table 1). Thus, the morphology of the *E. magnusii* cells mainly depended on the substrate and its concentration. The cells grown on 0.5% and 2% glucose looked more elongated and, on average, 1.85 times longer than those grown on glycerol (by cell medians) (Figure 2B,C,E). It should be noted that the formation of mycelial structures is not typical for *E. magnusii* species under the normal conditions [30]. Such a dimorphic transition could indicate a stress state of the culture induced by nutritional restriction.

Examining the survival rate of the culture during long-term cultivation, we showed that in one, two, or three weeks of growth, the culture utilizing 0.2% glucose showed an almost 1.5-fold higher reproductive capability than that using glycerol (Figure 3B). At the same time, most of the *E. magnusii* cells in all of the variants displayed a high survival rate by staining with vital dyes (Figure 3A) and PI (Appendix A). In the latter case, 98–99% of the cells showed negative staining throughout the whole experiment (Appendix A). Being stained with methylene blue, the cells using 2% glucose showed only 75% survival, while those grown upon nutritional restriction showed a nearly 100% survival rate (Figure 3A). These data show that calorie restriction significantly increased RLS. Extending life-span with CR is well known. Numerous genes concerning respiratory processes, gluconeogenesis, and peroxisomal functions in the yeast cells are repressed at high glucose concentrations [49]. Thus, ethanol fermentation is predominant in the cells grown at high concentrations of glucose, even in the presence of oxygen, mainly due to an excess of pyruvate, which is as a resulted of increased glycolysis, causing the generation of the mitochondria pyruvate dehydrogenase and blocking the respiratory process [50]. At the same time, both tricarboxylic acid cycle and respiratory genes as a result of heat shock and antioxidant genes are expressed if the glucose concentration decreases from 2% to 0.5% in the cultivation medium. This could indicate a shift towards respiration and stress tolerance if yeast populations achieve glucose levels similar to those used in caloric-restricted studies [51]. The correlation between glucose limitation and respiration shift agrees well with the fact that the CYT1-encoded cytochrome c1 together with glucose limitation is shown to prolongate RLS. However, the overexpression of the respiration transcription activator *HAP4* results in an increased life-span in 2% glucose media [11]. Moreover, metabolic transition to respiration has been proved to be essential for CLS expansion by glucose restriction, while the restriction of non-fermentable sugars prolonged no CLS in the yeast [52]. These data suggest that enhanced respiration is a life-prolonging mechanism in a specific form of CR, namely glucose restriction.

Our data partly support this hypothesis, demonstrating that the level of respiratory activity of *E. magnusii* cells grown with a low glucose concentration significantly exceeded that in the culture utilizing 2% carbohydrate at the logarithmic growth stage (Figure 4). Moreover, in four weeks of growth, the respiration rate of the cells using 0.2% glucose was twice as high as that using the 0.5% one, and approached the values obtained in the cultures grown without caloric restriction (Figure 4). The respiration rate is the most important feature of the energy status, and the induction of an alternative oxidase in respiration is considered to be one of the universal mechanisms of a cell’s response to stresses of various natures. The promotion of an alternative oxidase allows not only to avoid the electron “leakage” from the electron transport chain and thus to reduce the ROS generation, but also makes the process of respiration less energetically favorable [53]. However, under the conditions of caloric restriction, we observed no signs of activation of alternative mitochondrial oxidase upon inhibition of the cytochrome pathway by cyanide (Appendix A). This could indicate no stress pressure under those conditions and a high cell adaptation. These data confirm the hypothesis that CR may facilitate the ability to cope with oxidative stress [54]. The carbohydrate composition of fungi is an extremely important physiological and biochemical feature depending on the growth stage and its physiological state. Thus, the accumulation or consumption of certain carbohydrates may be associated with spore sporulation and germination, thermal or osmotic shock, and oxidative stress [55,56].

There is a large group of carbohydrates capable of performing a protective function under the conditions of stress. Among them, there are the disaccharide of trehalose, some polyols, namely glycerol, glycosylglycerol, arabitol, mannitol, and sorbitol. The metabolites have a high protective potential, providing protein stabilization under drought, osmotic shock, and high or low temperatures [57]. Metabolic rearrangement of the carbohydrate composition upon adaptation to CR revealed two main patterns: (1) extremely variable carbohydrate spectrum of the cells after a week of cultivation (Figure 5B), and (2) as the culture becomes older, a definite trend towards the accumulation of arabitol and inositol under the conditions without substrate restriction and at low glucose concentrations, respectively (Figure 5B). After a week of cultivation on glycerol as the substrate, in the cytosol carbohydrates composition, all the carbohydrates, with the exception of mannitol, were detected, while in the cells using 2% glucose, arabitol made up more than 90% of the total carbohydrate content (Figure 5B). Under caloric restriction of 0.5% glucose, the arabitol amount reached 83.7% Mannitol, inositol, glucose, and glycerol were detected as minorities (Figure 5B). In the culture growing on 0.2% glucose, in addition to all the above-mentioned components, trehalose (13.5%) appeared (Figure 5B). The natural disaccharide of trehalose, being a universal signaling and protective agent in fungal cells, is a powerful antioxidant and acts as a stabilizer of proteins and phospholipids in the lipid membrane bilayer. Thus, in the studies using *S. cerevisiae*, upon mild heat shock there was trehalose accumulation accompanied by an increase in survival rate [58]. At the same time, the *tps1tps2* mutant strains of *S. cerevisiae*, unable to synthesize trehalose, were more sensitive to ROS than the wild type. This could confirm the high antioxidant activity of trehalose in the yeast. The accumulation of trehalose performing a membrane protective function is likely an important adaptive element in CR in *E. magnusii* in the first stage of long-term cultivation. It is of interest that in all variants of calorie restriction, beginning from the first week of cultivation, a share of glycerol made up 1 to 7% (Figure 5B). The protective function of glycerol has been shown for many species of fungi. For example, under cold and osmotic stresses in *Aspergillus niger*, a representative of ascomycetes, the glycerol amount significantly increased [59]. The mannitol and glycerol levels increased upon hypothermy in the fungi [60]. It can be assumed that such a stable content of glycerol under severe CR conditions can play both protective and reserve functions.

Under CR, besides glycerol, the carbohydrate composition of *E. magnusii* cells contained a certain share of arabitol (from 2.5 to 13% at the late stages) (Figure 5B). Under the thermal, oxidative, and osmotic kinds of stress, the accumulation of *D*-arabitol, together with trehalose, was also shown for the *Debaryomyces* and *Geotrichum* fungi [61], the yeast of *Kluyveromyces lactis* [62], and *Candida albicans* [55]. Arabitol is known to be a by-product of the pentose phosphate pathway, which is needed in order for eukaryotic cells to replenish the pool of reducing equivalents of NADPH [61]. This property brings this polyol closer to mannitol, which, being oxidized to fructose, can also reduce NADP to NADPH [61]. NADPH, in turn, is needed to restore the oxidized glutathione, a universal molecule maintaining the redox potential in the cell [63]. The appearance of a small amount of mannitol in the cells using 0.2% glucose at the four-week growth stage also proved to be rather interesting. Mannitol is a sugar alcohol, non-cyclic hexitol, and is the most wide spread natural polyol that performs the most crucial physiological functions in a fungal cell, such as the storage of carbohydrates, maintaining the balance of reducing equivalents, osmoregulation, modulation of coenzymes, regulation of cellular pH due to proton donation, and resistance to abiotic stressors [64].

The mechanisms of the protective action of mannitol are still being discussed. However, there are two main hypothetical opinions: according to one of them, mannitol acts as a so-called compatible compound, which accumulates in high concentrations under stress and is combined with normal cellular functions; according to the second assumption, mannitol acts only as an antioxidant [65]. Its appearance in the aging culture at the late stages can also possibly play a protective role.

It should also be noted that, nowadays, natural polyols are in-demand compounds in food, pharmaceutical, and medical industries. Thus, six-carbon polyalcohol of mannitol has wide applications in biotechnology, serving as a sweetener (1.6 kcal per 1 g), texturing, and conserving agent of food products [66]. Thus, the capability of some fungi effectively producing carbohydrates under the selected conditions has been widely used in industry. The new *Candida parapsilosis* strain SK26.001 was designed as a potential source of mannitol, producing up to 97.1 g/L of polyol under the optimal conditions at a high glucose concentration [67]. This possibility was also shown for the poly-extremophilic *Y. lipolityca* ATCC8661, which, using industrial glycerol, can synthesize beside mannitol erythritol [68]. Arabitol, being in the pentite family, can be used as a sweetener in the food industry, an anticaries agent and adipose tissue reducer in the medicine industry. It can also serve as a substrate for synthesizing compounds such as arabic and xylonic acids, propylene, ethylene glycol, xylitol, and some others. Some strains of the *Candida* and *Pichia* genera are among the most important producers of arabitol. It is included in the list of 12 structural blocks of C3–C6 compounds, which are of great interest for bioconversion or for the biotransformation of agricultural wastes for the forest industry (1-arabinose and glucose) and biodiesel industry (glycerol) [69]. Some organisms can convert glucose-6-phosphate to *D*-ribulose-5-phosphate, dephosphorylating the latter, and then reduce *D*-ribulose to *D*-arabitol using NADP-dependent *D*-arabitol dehydrogenase. Moreover, in a yeast cell, arabitol can be synthesized from glycerol via the same pathways as it is from glucose via glucose-6-phosphate [69].

Hence, by varying the concentrations of the initial growth substrates, we can obtain some special products using various concentrations of substrates and some other conditions. This allows us to successfully use DR in yeast-based biotechnological processes.

## 5. Conclusions Remarks

The cellular metabolism is an essential factor of aging and longevity for eukariotic or-ganisms. Some of its shifts observed during aging may be regarded as metabolic bi-omarkers typical for aging changes, such as deterioration in cellular functionality, tissues and organs, homeostasis, and organismal health [70,71,72]. Moreover, the aging hallmarks contain nuclear DNA damage and repair, alterations of epigenetic regulation, telomere shortening, proteo-toxic stress, nutrient sensing deregulation, deterioration of the mitochon-dria, cellular senescence, a decrease in stem cell number, and a distortion of intercellular communications [73,74]. Using all of the findings, the appearance of delayed aging due to the so-called metabolic “clock”, “signature”, “trace”, or “profile” could be proposed [73,74]. According to our data, some markers crucial for the adaptation of yeast to long-lasting cultivation under both the conditions of caloric restriction and without nutrients limitation could be applied to the unicellular eukaryotes. First, it is essential to remodel the carbohydrate cytosolic composition, in order to provide a stable state of cellular membrane apparatus and the action of the energy mitochondria system, as well as the changes in the cellular morphology coupled with metabolic adaptation. Furthermore, CR is supposed to play a significant role in carbohydrate metabolism regulation in the yeast cell. It can be considered as an important agent in the regulation of biotechnological processes using yeast-based polyalcohol producers.

## Figures and Tables

**Figure 1 microorganisms-10-01709-f001:**
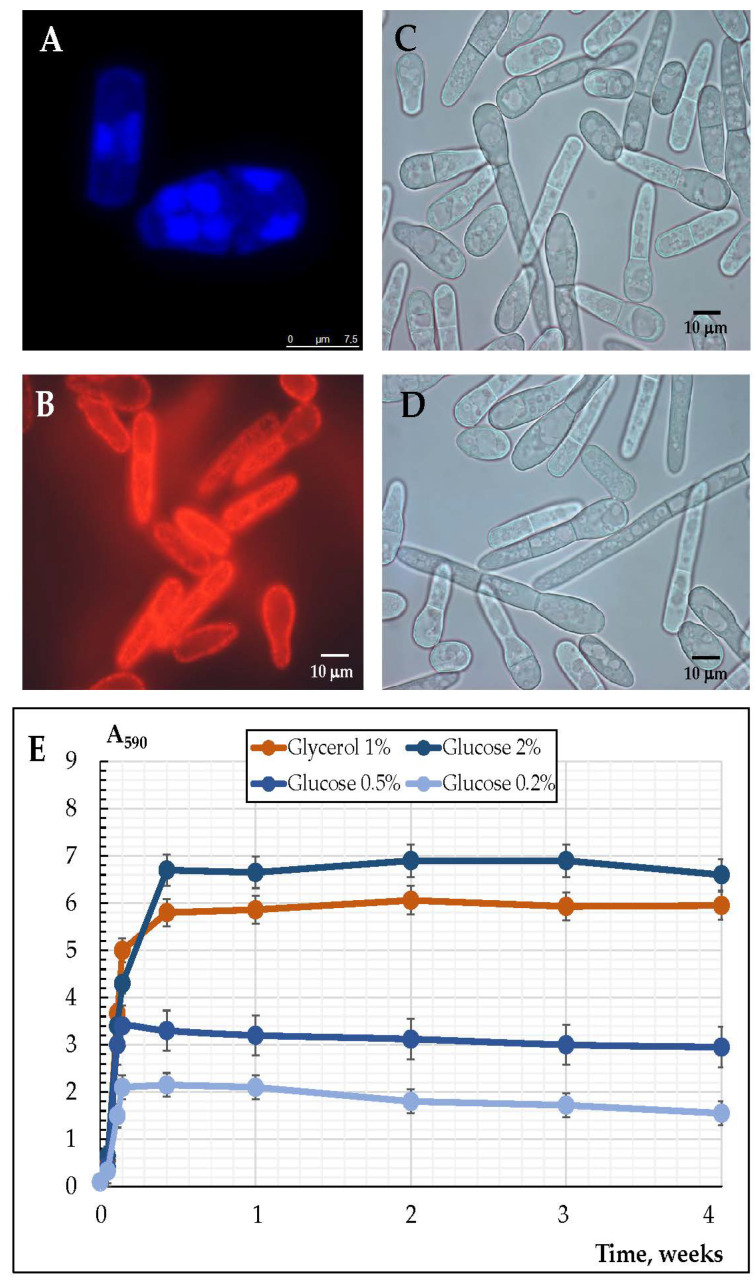
Micro-images (**A**–**D**) and growth curves (**E**) of the *E. magnusii* cells using various substrates. (**A**) The cells were increased in the logarithmic stage and labeled with 0.3 μM DAPI for DNA. The yeast cells were resuspended in PBS-containing DAPI (50 μg per mL), mixed gently, and incubated in the dark at room temperature for 15 min. The cells were visualized with a fluorescence microscope (at 100×) using a blue UV filter. (**B**) Potential-dependent staining of mitochondria in the *E. magnusii* cells raised in the logarithmic growth phase with Rh123. The cells were incubated with Rh123 for 20 min. The incubation medium contained 0.01 M PBS, 1% glycerol, pH 7.4. The areas of high mitochondrial polarization are indicated by bright-red fluorescence because of the concentrated dye. To examine the Rh123-stained preparations, filters of 02 and 15 (Zeiss) were used (magnification 100×). (**C**,**D**) Pseudomycelial and true mycelial forms of cells. (**E**) Growth curves of the *E. magnusii* yeast during long-lasting cultivation using different substrates.

**Figure 2 microorganisms-10-01709-f002:**
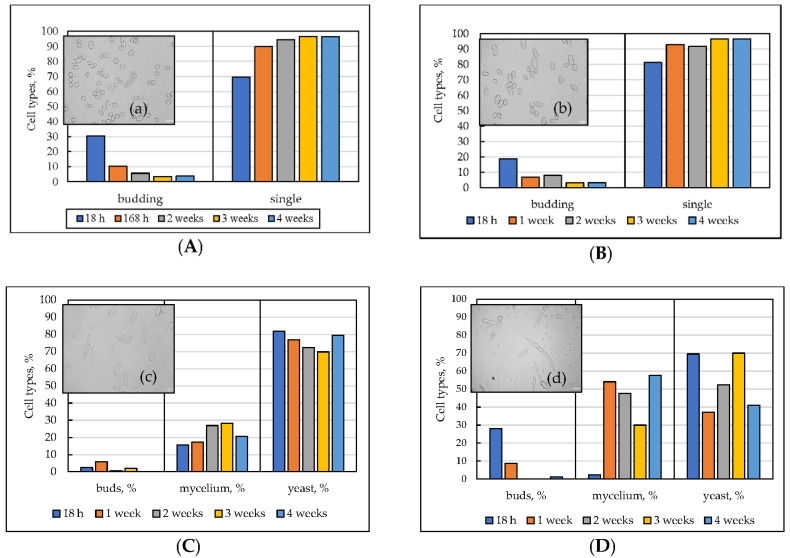
Micro images (**a**–**d**), distribution of cell morpho-types (**A**–**D**), and morphometric parameters (**E**,**F**) of the *E. magnusii* cells using different substrates during long-lasting cultivation. The photos were taken with an AxioCam MRc camera (magnification 100×): **A**(**a**) glycerol; **B**(**b**) 2% glucose; **C**(**c**) 0.5% glucose; (**D**(**d**) 0.2% glucose. (**E**) Bar chart of the changes in the median values of cell length; (**F**) bar-chart of the changes in the median values of the cell width.

**Figure 3 microorganisms-10-01709-f003:**
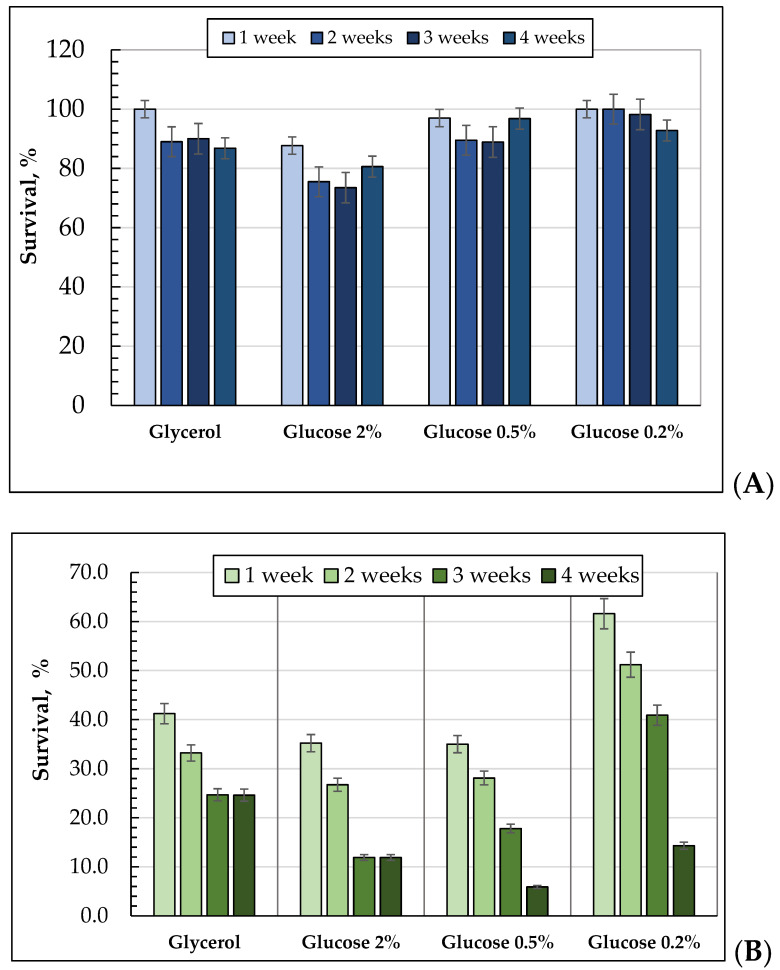
The survival rate of *E. magnusii* yeast grown on different substrates during long-term cultivation using the methods of methylene blue (**A**) and colony-forming units (CFU) (**B**).

**Figure 4 microorganisms-10-01709-f004:**
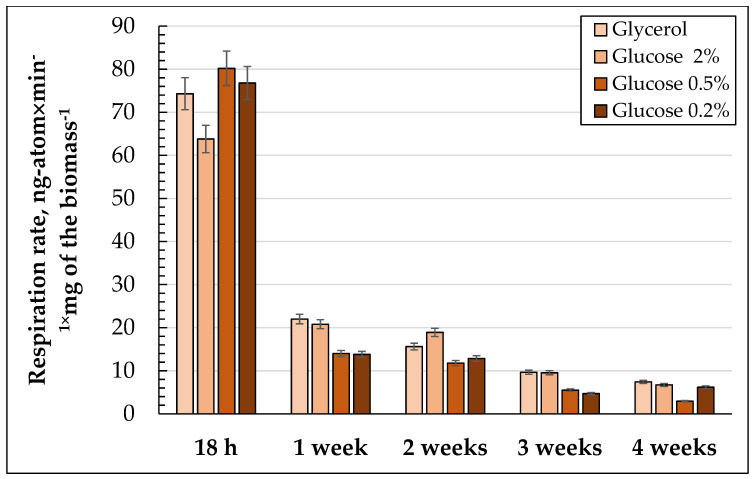
Respiratory activity of *E. magnusii* yeast during long-lasting cultivation using different substrates.

**Figure 5 microorganisms-10-01709-f005:**
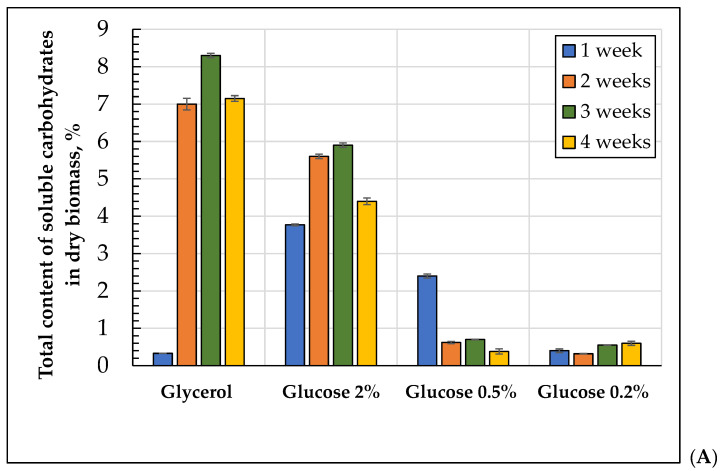
Changes in the total (**A**) and relative (**B**) content of soluble cytosolic carbohydrates of the *E. magnusii* cells (% of dry biomass) during the long-term cultivation using different substrates.

**Table 1 microorganisms-10-01709-t001:** Results of two-way analysis of variance with repetitions (*n* = 100).

Source of Changes in the Cells Length	SS	df	MS	F	*p*-Value	F Critical
**Substrate used**	30,285.72	3	10,095.24	396.44	1.20 × 10^−201^	2.61
**Age of the culture**	4604.02	4	1151.01	45.20	2.26 × 10^−36^	2.38
**Interaction of the factors**	2139.72	12	178.31	7.00	1.37 × 10^−12^	1.76
**Overlooked effects**	50,419.62	1980	25.47	-	-	-
**Total**	87,449.089	1999	-	-	-	-

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
