# Peer review of "The Effect of Different Substrates on the Morphological Features and Polyols Production of Endomyces magnusii Yeast during Long-Lasting Cultivation"

_microorganisms, 2022, doi:10.3390/microorganisms10091709_

Round 1

Reviewer 1 Report

General Comment:

The study investigated the effects of calorie restriction on the biochemical and morphometric parameters of Endomyces magnusii yeast during growth on glycerol and glucose substrates. The manuscript contains many grammatical and spelling errors that need to be addressed before the manuscript is accepted. In addition, the author should address the following concerns.

11.     Lines 216-224: Why call glycerol an oxidative substrate and glucose a ‘fermentative’ one when glucose is more oxidized when compared to glycerol?

22.     Figure 3: Why was there a stark difference between methylene blue and colony forming units (CFU) methods of measuring survival rate? The CFU method appears to be a better method of determining survival rate.

33.     Lines 113 – 119: I don’t agree with the conclusion on why CFU method gave a lower survival rate when compared to methylene blue or PI method. If the cells were quiescent or dormant due to calorie restriction as claimed by the authors, the transfer of the dormant cells to YPD medium should have provided nutrients for the supposedly quiescent cells to flourish on. This means that the quiescent cells should have grown when plated on YPD agar. It could be that the supposedly quiescent cells were in an early apoptotic stage where their cell membranes are still intact and were not recognized as dead cells by the methylene blue method and failed to grow when plated on YPD agar.

44.     Lines 137 – 139: The sentence in this section is hard to comprehend. This type of grammatical errors that make understanding difficult abounds throughout the manuscript. The second sentence in the paragraph reads ‘…the respiratory rate decreased and gradually declined…’ The words ‘decrease’ and ‘declined’ means same thing and the authors should use one and strike out the other.

55.     Lines 179- 180: Please add appropriate references where soluble inositol was reported in Saccharomyces cerevisiae.

66.     Line 193: The manuscript has a ‘Results and discussion’ section and then a separate ‘Discussion’ section.

77.     Line 251: The authors used the term ‘DR’ many times in the discussion section of the manuscript without giving its full meaning. I assumed the authors were referring to DR as diet restriction. The authors must be consistent with terms used in the manuscript so as not to alienate readers. In the introductory section of the manuscript, the authors used CR – calorie restriction, but then switched to DR in the discussion section.

Author Response

Dear Reviewer!

Our comments are in attached file

Reviewer 2 Report

Dear editor

The paper proposed by Kokoreva present interesting results about the influence of different concentrations of glucose (2%, 0.5%, and 0.2%) and glycerol on the morphological features and polyols production of the Endomyces magnusii yeast during long-lasting cultivation. The work is well presented and structured. The paper could be read by a native speaker to ensure the correct use of English in few points, however there are no major grammar mistakes that would make the text difficult to comprehend.

Author Response

Dear Reviewer!

Our comments are in attached file.

Reviewer 3 Report

In this article, the authors endeavour to study how tha quality and/or quality of the carbon source influences long term viability as well as cell morphology. To achieve this, they use various techniques such as flow cytometry or cell staining as well as statistical analysis to support their conclusions. These are usually very well thought and carried out.

Overall, the reading is difficult due to the authors lack of fluency in English. they should ask an English-speaking science writer to improve this text.

In the introduction, the author fail to mention why they have chosen this particular yeast species besides the fact that they have already published a study on it. This species being not well known at the moment requires that the authors indicate if it has a special primary metabolism or that it has some kind of industrial application warranting this study since as they say in their introduction other yeast models have been used for a long time. Although very interesting, their paragraph on the mechanisms are besides the point in this part since the study is a phenotypic analysis of this species. It is however of interest in the discussion to interpret the results and put them in context. It is not clear either if this strict aerobe ferments as well as respirates or only respirates, an information required to better understand the results obtained. I would like also to point out in relation to the previous remark that if it ferments then their is a diauxic shift between glucose and ethanol but if it only respirates then their is no diauxic shift but a biphasic growth due to the adaptation to limiting glucose concentrations at the end of the growth. Furthermore, one cannot describe stationary phase as a growth phase since stationary means no movement and in its late stages is quiescence.

In the material and methods, the authors mention no provenance of their strain or if it is a wild type or not.

In the results, it is important to keep in mind what the different phases of the cultures are to better describe the results as well as if the cells only respirate or ferment also. In part 3.1 the authors refer to hours but in the corresponding graph it is weeks that are used, it would be good to make sure both the same. As a remark here, I think it is important to keep in mind that 1% glycerol has the same carbon content as 0.5% glucose and it would be fit the authors to mention whether glycerol is converted to glycerol 3-phosphate in same manner as in S. cerevisiae or not. I would also eliminate lines 219-222 which make reference to another experiment about cell morphology and not cell growth, it is confusing. The authors could shorten this part by saying that on glucose 2% growth has 3 phases: exponential, a slower one and stationary phase and give the time of each phase and this for the 3 other conditions. And then describe the biomass yield in another paragraph.

3.2, the authors should highlight the fact this yeast does not display similar cell morpholgies between 1% glycerol and 0.5% glucose despite being carbon equivalents. I am not very convinced by the description budding cells but I must admit that I have as budding reference S. cerevisiae, could the authors select a field in which budding is clear? Or is cell division occuring through septation? Here the authors make histograms with error bars but never assess whether the differences are statiscally significant or not despite having the know-how to do so i figures 2E and F. Page 12 line 5, I think the authors mean less than 0.005?

3.3 Methylene blue is a viability marker, i.e. are cellular integrity, as is PI so these results should be similar. Growth on plates is for me the true viability test and should be comparable or lower than the previous 2 tests. In this regard the viability test is very surprising to me and counter-intuitive.

I find figures 4-6 very difficult to analyze and in my opinion they should be put in the supplementary data and a histogram should put all these results in a more readable format. I agree with the partioning the authors propose for most cell distribution, I don't agree with the one shown in 5B, to me the population is unique but spread more widely, which would be more consistent with the other results the authors have shown.

Figure 7, I don't see the point of the panel B. Exponentially grown cells are almost totally inhibited by cyanide and after 4 weeks, 67% of the remaining respiring cells (<10%) are inhibited by cyanide, which comes out the same, almost all cells are poisoned by cyanide.

Figure 8 is the most surprising result with the switch from accumulating mainly arabitol in the first week when cells are grown on 0.5% glucose to inositol the following weeks. Is arabitol metabolised or converted into inositol?

As a curiosity, what do the authors think the results would be if they would have used 4% glycerol, which is equivalent to 2% glucose. In my opinion the results would be similar to those obtained with 2% glycerol, suggesting that it is a transport issue more than a caloric restriction issue based on the plethora of glucose transporters with very wide ranging Km. 

Finally, the authors should shorten the conclusion to make more to the point.

Author Response

(The authors gave the same response as above.)
